# Research Progress of Betalain in Response to Adverse Stresses and Evolutionary Relationship Compared with Anthocyanin

**DOI:** 10.3390/molecules24173078

**Published:** 2019-08-24

**Authors:** Ge Li, Xiaoqing Meng, Mingku Zhu, Zongyun Li

**Affiliations:** School of Life Science, Jiangsu Key laboratory of Phylogenomics & Comparative Genomics, Jiangsu Normal University, Xuzhou 221116, Jiangsu, China

**Keywords:** anthocyanins, betalains, caryophyllales, evolution, pigment biosynthesis, stress tolerance

## Abstract

Betalains are applicable to many aspects of life, and their properties, characteristics, extraction and biosynthesis process have been thoroughly studied. Although betalains are functionally similar to anthocyanins and can substitute for them to provide pigments for plant color, it is rare to study the roles of betalains in plant responses to adverse environmental conditions. Owing to their antioxidant capability to remove excess reactive oxygen species (ROS) in plants and humans, betalains have attracted much attention due to their bioactivity. In addition, betalains can also act as osmotic substances to regulate osmotic pressure in plants and play important roles in plant responses to adverse environmental conditions. The study of the physiological evolution of betalains is almost complete but remains complicated because the evolutionary relationship between betalains and anthocyanins is still uncertain. In this review, to provide a reference for the in-depth study of betalains compared with anthocyanins, the biochemical properties, biosynthesis process and roles of betalains in response to environmental stress are reviewed, and the relationship between betalains and anthocyanins is discussed.

## 1. Introduction

The predominant plant pigments in nature are carotenoids and anthocyanins, which are produced in almost all kinds of plants except for some species in Caryophyllales [1]. Recently, owing to their antioxidant capability to remove excess reactive oxygen species (ROS) in plants and humans, betalains have attracted much attention due to their bioactivities [2]. However, the biosynthesis and regulatory pathways of betalains are currently only partially understood, their origins are uncertain, and there is a complex evolutionary relationship between betalains and anthocyanins, as both compounds are functionally replaceable but mutually exclusive and noncoexistent in natural plants.

### 1.1. Basic Information Discerning Betalains and Anthocyanins

Betalains are found in some plant species of the Caryophyllales, such as *Celosia argentea* L. (inflorescences), *Beta vulgaris* L. (whole plants), *Opuntia ficus-indica* L. (fruits), *Bougainvillea glabra* (bracts), and *Portulaca oleracea* L. and *Mirabilis jalapa* L. (flowers) [3]. Betalains are tyrosine-derived, water-soluble, nitrogen-containing pigments and can be divided into two types: red-violet betacyanins and yellow betaxanthins. Benefiting from their antioxidant and anti-inflammatory abilities, both have high nutritional value and positive effects on health and disease [3,4,5]. Betalains were once thought to be the same compounds as anthocyanidins found in beet plants because they appear to replace anthocyanins and to perform the same functions as anthocyanins do in other plant species, such as providing color to flowers and fruits and responding to environmental stress [1,2,4,6]. In addition, betalains have also been detected in some higher fungi [7], and recently, some researchers identified the first bacterium able to synthesize betalains [8]; however, the view that the “first betalain-producing bacteria break the exclusive presence of the pigments in the plant kingdom” [8] may be not right, betalain also exists in fungi. Similar to anthocyanins, betalains are most often noticeable in the petals of flowers, and they are also found in the fruits, leaves, stems and roots of plants (Figure 1). Both anthocyanins and betalains are stored as glycosides in cell vacuoles and share similar histological locations in dermal and vascular tissues of vegetative organs [6].

### 1.2. Identification and Detection of Betalains 

Betalains can be identified simply by their change in color in different acidic/basic solutions. These compounds are red-violet in water and stable in the pH range 4.0–7.0; when the pH < 4.0 or > 7.0, the color of the solution changes from red to purple, and when the pH > 10.0, the color of the solution rapidly turns yellow [9]. The color change of solutions of betalain extracted from beet roots with three different pH values is shown in Figure 2. In addition, betalains can also be identified by their absorbance. The absorbance can be used to calculate the content of betalains (BC) using the following modified equation [10]: BC [μg g^−1^] = [(A_538_ or A_465_ × DF × MW × 1000)/(ε × L)]. Here, A_538_ and A_465_ represent the absorbance of betacyanin and betaxanthin, respectively; DF is the dilution factor; MW is the molecular weight (550 g mol^−1^ for betanin and 308 g mol^−1^ for indicaxanthin); ε is the molar extinction coefficient (60,000 L mol^−1^·cm^−1^ for betanin and 48,000 L mol^−1^·cm^−1^ for indicaxanthin) [11,12]; and L is the path length of the 1 cm cuvette. In addition, the identification and detection of betalains can be performed using more advanced techniques, e.g., reverse phase high-performance liquid chromatography (RP-HPLC) and positive ion electrospray mass spectrometry [13,14], HPLC- diode-array detector (DAD), HPLC-MS, HPLC-electrospray ionization mass spectrometry (ESI-MS) and NMR techniques [15,16,17]. These methods can provide some data support for changes in the biosynthesis pathway of betalain, such as different content of betanin and phyllocactin detected in callus lines from colored quinoa varieties (*Chenopodium quinoa* Willd) indicates that the biosynthesis pathway of betalain has been altered with the main accumulation of the intermediate instead of the final compound [15].

### 1.3. The Roles of Betalains in Humans and Plants

As important plant secondary metabolites, betalains have strong antioxidant properties and can scavenge oxygen free radicals directly or indirectly both in vivo and in vitro. These metabolites have been used to treat human dysfunction related to peroxidative stress [18,19]. The oxidative/antioxidative mechanisms of the human body are involved in the pathological processes of many conditions, such as tumors, cancer, arteriosclerosis, diabetes and aging [4], which are mostly associated with the excessive production of ROS [19]. There are some common excellent vegetables and fruits, such as *Spinacia oleracea* L., *Amaranthus mangostanus* L., *Hylocereus undulatus Britt.*, which all belong to Caryophyllales, and contain highly concentrated betalains instead of anthocyanins, thus indicating that eating these vegetables and fruits is very beneficial to human health. There are also some common horticultural flower plants, such as *Celosia cristata*, *Dianthus caryophyllus* L. which enrich and decorate human life. Moreover, some Caryophyllales plants can be medicinal and may benefit from the function of betalain, such as *Phytolacca esculenta* (root) [20].

Previous studies have shown that ROS are also integral components of plant responses to stress [6]. Therefore, betalains can also be used to assist plants in responding to stress by eliminating ROS. For example, ROS can induce betalain biosynthesis in red beet leaves when it suffer from wounding and bacterial infiltration that may occur through neutralization reactions [21,22]. Actually, red beets have been proven for antimicrobial applications [7,23,24]. There is a study specifically reporting that betanin, a type of betalain, has exceptionally high free radical-scavenging activity, and at a pH > 4, betanin is approximately 1.5–2.0-fold more active than some anthocyanins which are considered pretty good free radical scavengers, as determined with Trolox equivalent antioxidant capacity (TEAC) assays [25]. Many species of Caryophyllales are sand-fixing plants in arid areas, of which *Haloxylon ammodendron* is a very important afforestation species in the sand area.

### 1.4. The Significance of Betalains and Anthocyanins in Plants

Betalains and anthocyanins can be used as ROS scavengers to remove ROS directly or indirectly and can maintain homeostasis as an osmotic substance that regulates cell osmotic pressure to assist plants in adapting to harsh external environments [25,26,27]. In some severe saline-alkali and/or arid areas, some unique species of Caryophyllales are present, such as *Opuntia stricta* in the desert and *Suaeda salsa* in coastal salt flats. The majority of red-leafed plants grow in marginal environments [6]. And recent a study reported that the abundance of betalain was 2–5-fold higher in the red leaf sectors than in the green leaf in *Amaranthus tricolor* L. [28]. Therefore, the potential application and development value of betalains and anthocyanins have very promising prospects, and some betalain/anthocyanin synthesis-related genetic material could be used to cope with future climate change. However, the evolutionary relationship between betalains and anthocyanins is still uncertain, although the significance and reason why these two compounds are functionally replaceable but mutually exclusive and noncoexistent in nature remain unknown.

## 2. Comprehensive Effects of Plants in Response to Various Stresses

Plants in natural environments inevitably suffer from various types of stress, such as extreme temperature, drought, and salinity. After the appearance of oxygen-emitting photosynthetic organisms approximately 250 million years ago, ROS have accompanied the aerobic metabolism of life on Earth as an inevitable byproduct [6,29]. Excessive ROS will cause oxidative stress in plants. ROS can destroy the membranes of cells and organelles, denature proteins and damage nucleic acids (DNA or RNA), affecting the normal biological processes of plants and even causing death in severe cases [30]. The real magic is that during the long-term evolutionary process, plants acquired an ROS-scavenging system that protects them from ROS, the ROS-scavenging system (Figure 3) can alleviate the above damaging process [31,32]. There are two types of antioxidant systems in living organisms: an enzymatic antioxidant system, e.g., superoxide dismutase (SOD), and a nonenzymatic antioxidant system, e.g., vitamin C, anthocyanin, betalain, catechin [6,22,26].

In fact, substantial changes will be induced when plants are exposed to adverse environments, such as unfavorable osmotic pressure [33] and ionic homeostasis [34,35]. Therefore, there are some substances such as proline that can be important osmotic adjustment substances that respond to the abovementioned changes. The functional verification of stress tolerance-related genes is usually conducted by measuring the amount of proline or by quantitatively detecting the expression of genes involved in proline biosynthesis, such as *P5CS1* and *P5CS2* [36,37,38]. Surprisingly, it has been found that betalain not only exerts effects similar to those of anthocyanins and catechins and is an excellent scavenger of a great number of ROS types [6,39,40] but also works like proline to adjust the osmotic pressure of plants [34,39,41], anthocyanins do the same under similar conditions [42].

## 3. Functions of Anthocyanins and Betalains in Plant Responses to Adverse Environmental Conditions

In addition to providing pigmentation to assist plant reproduction and/or being a protective or aposematic coloration that protects plants from herbivores [43], both anthocyanin and betalain metabolites play important roles in plant abiotic and biotic stress resistance. The functions of anthocyanins in plants in response to adverse environmental conditions have been studied for many years. The possible benefits of betalains for stress tolerance are also gradually being increasingly discovered, which represents another challenging but interesting field.

### 3.1. Flavonoids and Anthocyanins in Plant Responses to Adverse Environmental Conditions

Flavonoids are ubiquitous in land plants. One of the most important flavonoids in plants is anthocyanin, which has been indicated to provide assistance to plants in tolerating extensive stress [44], such as cold [45,46], high light [47,48,49], weak light [49,50], oxidative stress [51,52], nutrient deprivation [18], UV light [53,54], drought [52,55], salinity [36,48,56,57,58], metal toxicity [59,60], and pest [61] and pathogen attack [46,62,63], all showed increases in the anthocyanin content when the plant responded to the above various stresses. For example, with the salt concentration increases from 0 to 150 mmol·L^−1^, the anthocyanin content increases from about 16 to 20 mg·g^−1^ DW in the leaves of *Carthamus tinctorius* var [36]. The diverse functions of anthocyanins can be summarized as follows: they (1) intercept and absorb light energy to avoid photodamage caused by excess light energy under abiotic environmental stress, (2) protect antioxidant enzymes, (3) directly scavenge free radicals, (4) indirectly remove ROS by interacting with other molecules in other signaling pathways, and so on [31,64]. Previous studies have shown that anthocyanins play a role in plant responses to adverse environmental conditions mainly via aposematic signaling, camouflage, mimicry of unpalatable foliage or undermining the crypsis of herbivorous insects [31,43,65,66,67]. These findings indicated that supplemental anthocyanins exogenously added to *Arabidopsis* leaves showed a protective effect under high light, and the total anthocyanin content of purple sweet potato obtained by solid-phase extraction was 66 mg·g^−1^ [64]. For *Raphanus sativus*, the anthocyanin levels in Man Tang Hong flesh and Hong Feng No. 1 skin were found to be 4.69 and 4.39 mg·g^−1^ dry weight, respectively; although the levels of these anthocyanin compounds and contents were similar, the expression patterns of related genes differed [68], and these small differences may give different *Raphanus sativus* cultivars different resistance capabilities to adverse environmental conditions.

### 3.2. Betalains in Plant Response to Adverse Environmental Conditions

Increasing evidence suggests that betalains play critical roles in protecting plants against various adverse stress conditions. Jain et al. [26] clarified the functional significance of betalain biosynthesis in leaves of *Disphyma australe* under salinity stress, which apparently granted photoprotection to chloroplasts suffering from salinity stress. The roles and relationships between betalains and anthocyanins in salinity and drought stress in plants have been characterized [6]. Similarly, one paper showed that there was a decline in chlorophyll content and a 4-fold increase in betalain concentration as a consequence of salinity stress in the halophyte *Salicornia fruticosa* [69]. For tolerance to saline stress in *Portulaca oleracea* L., research shows some increase in proline content and betalain pigmentation and in *PC5S* gene expression coding enzyme reliable to proline biosynthesis [70]. The function of betalain in the drought tolerance of some plant species has already been documented [34,71]; this function, together with light [72,73], UV radiation [71,73], low temperatures [72], etc. is an environmental cue that induces betalain production [6]. It has been indicated that both betacyanins and anthocyanins are practical photoprotectants and are efficient scavengers of ROS in plants facing a variety of abiotic stressors [6,72], which is consistent with the roles shown in Figure 3.

Increasing betacyanin concentrations in response to different stresses have been observed in other species of the Caryophyllales that produce betalains [7,21]. It was reported that in nontreated red beet leaves, the betacyanin content (Bc) was 4 μg·disc^−1^, while in wounded leaves, the betacyanin levels increased to 28 μg·disc^−1^ at 48 h [21]. In some anthocyanin-producing species, it has been reported that the genes of the betalain biosynthesis pathway are heterologously overexpressed, which can produce high levels of betalains, for example, tobacco leaf tissue with a betalain content of 135 mg·kg^−1^ fresh weight, tomato fruit with a betalain content of 200 mg·kg^−1^ and eggplant fruit with 120 mg·kg^−1^ [74]. These betalain-producing transgenic plants exhibited enhanced plant tolerance to stress [74]. In *Beta vulgaris* L., it was reported that total betalain concentration increased in a dose dependent manner by 8%, 28% and 34% respectively compared to the control in three different doses of UV-B radiation (3.042, 6.084 and 9.126 kJm^−2^d^−1^) [75]. A previous study showed a seven-fold increase in the production of tyrosine-derived betalain pigments by manipulation of tyrosine availability, with an upper range of 855 mg·kg^−1^·fresh weight [76], which gives us a direction to improve the resistance of betalain-producing plants. A recent study reported that cysteine can enhance the content of betalains and polyphenols in red beet and showed that betalains contribute more than phenolics to the antioxidant capacity [77]. The above findings illustrate that betalain is potent at enhancing plant stress tolerance.

## 4. Comparison of the Biosynthesis Pathways of Betalains and Anthocyanins

Using scientific methods such as transcriptome analysis, researchers have gradually explored the important enzymes and genes (shown in Figure 4 and Figure 5) of these two types of pigment biosynthesis pathways [78]. To better determine the differences between the two synthetic pathways, we mapped the biosynthesis pathways of betalains and anthocyanins in accordance with previous studies; the results are shown in Figure 4 and Figure 5 (adapted from [1,2,4,78,79,80,81]), respectively. Interestingly, betalains and anthocyanins are mutually exclusive in the same plant [1,79,82]; in Figure 6, it can clearly be seen that the starting substrates for betalain and anthocyanin synthesis are different, but both tyrosine and phenylalanine are aromatic amino acids and share the same branched acid synthesis pathway. Prephenic acid, commonly known by its anionic form prephenate, is an intermediate in the biosynthesis of the aromatic amino acids phenylalanine and tyrosine. Moreover, there is metabolic synergism between tyrosine and phenylalanine [83,84] as well as some feedback regulation during the biosynthesis process; for example, it is possible that L-Tyr inhibits chorismate mutase and that L-Phe inhibits prephenate dehydratase and, to a much greater extent, prephenate dehydrogenase [85]. However, the reason why these two types of pigments do not coexist remains unclear. In general, there is still debate regarding the origin of betalain within the core Caryophyllales and its mutual exclusion with anthocyanin in the realms of taxonomy, systematics, phytochemistry and evolutionary biology [1]. Surprisingly, there have been experiments to prove that it is possible to produce betalain in anthocyanin-producing plant species by providing an enzyme and substrate, such as 5,6-dihydroxy-phenylalanine (DOPA) dioxygenase and L-DOPA [82]. In addition, there are some documents demonstrating the production of betalain in anthocyanin-producing plants given only two enzymes, DOPA 4,5-dioxygenase (DOD) and cytochrome P450 (CYP450, e.g., CYP76AD1, CYP76AD5 and CYP76AD6) [2,27,81], and the processes can be regulated by v-myb avian myeloblastosis viral oncogene homolog (MYB) transcription factors; for example, BvMYB1/Y can upregulate the expression of CYP76AD1/R and BvDODA1 in beet [27]. Furthermore, phylogenetic analysis of BvMYB1 has clearly shown that it belongs to the anthocyanin MYB–bHLH–WD repeat (MBW) complex MYB clade (S6), distinguishing it from MBW-independent MYB regulators [2], such as AtMYB11, AtMYB12, and AtMYB111, all of which are independently capable of activating the genes that regulate early steps of the flavonoid pathway [86]. However, BvMYB1 cannot upregulate the anthocyanin pathway in *Arabidopsis* and has lost the ability to interact with the bHLH members of the MBW complex [2]; these phenomena may be due to the inability to form the MYB cofactor that causes the mutual exclusivity of betalains and anthocyanins.

Brockington et al. [1] reviewed the complex evolutionary relationship between betalains and anthocyanins and attempted to explain some of the main hypotheses proposed by researchers via phylogenic, comparative genetic and genomic data and suggested additional possible functional mechanisms. However, the reasons for the existence of three phenomena [1] “(1) the unique derived origin of betalain pigmentation, (2) the presence of both anthocyanin-pigmented and betalain-pigmented lineages in the core Caryophyllales, and (3) the mutual exclusivity of the two pigment types” are still not clearly explained. A comparative genetic study proposed possible mechanisms underlying the switches between pigment types and suggested that transcriptional downregulation of late-acting enzymes is responsible for a loss of anthocyanins [1].

In another paper, it was shown that anthocyanins could not be synthesized in *Mirabilis jalapa* L. because the anthocyanidin synthase (ANS), the key enzyme in the anthocyanin biosynthesis pathway, was truncated compared to ANS in species that normally produce anthocyanins; this was shown via transcriptome and metabolome analyses and via quantitative expression of the genes involved in the anthocyanin biosynthesis pathway [79]. The paper also speculated that the abovementioned *MjANS* truncation may be a unique example or a common one in species that produce betalains [79]; therefore, further research is needed.

Other researchers have also conducted comprehensive and objective reviews on the biosynthesis, source and application of betalain pigments, especially on betalain biosynthesis [3,4,6,87,88], but excepting the aspect of assisting plant response to stresses and evolutionary relationship between betalains and anthocyanins, which provide a broad perspective on the currently used and potential new sources for betalains, including the use of natural resources or metabolic engineering and summing up the potential applications of betalains in research and business applications [4,23,62,89].

## 5. Evolutionary Relationship between Betalains and Anthocyanins

Previous studies on the nature, characteristics, applications and biosynthesis of betalains and anthocyanins have been detailed and thorough. However, the two pigments do not exist in the same plant, and there are only a few species among the Caryophyllales that can produce betalains; this phenomenon is still confusing. Many researchers have been trying to prove whether anthocyanins or betalains are evolutionarily younger, revealing, for instance, key genes related to betalain biosynthesis and providing insight into betalain evolution by transcriptome and metabolome analyses [78,79]. These attempts further provide a possible explanation for the lack of anthocyanins in *M. jalapa* flowers by expounding *MjANS*, which encodes a key enzyme that is involved in anthocyanin biosynthesis but is highly expressed in betalain-accumulating *M. jalapa* flowers, with a major sequence deletion [79]. Comparisons of the stress tolerance of plants in harsh environments, such as the predominance of red-leafed plants in marginal environments [6], means that betalain-producing plants are more suitable for survival in those harsh environments. Even though it is difficult to know which one is evolutionary younger, that which one is applicable to the future is predictable. Moreover, a recent study showed that betalains contribute more antioxidant capacity than do phenolics [77]. By conducting evolved statistical or transcriptome sampling analysis, the phylogram from parsimony analysis of combined *rbcL*/*matK* plastid genes from 82 Caryophyllid taxa [1] and Caryophyllales phylogenomic analysis of 308 transcriptomes [90] provide some important references to our understanding of pigments evolution in Caryophyllales, indicating that the ancestral type was almost certainly anthocyanin [6].

Based on previous studies and our understanding, three possible mechanisms for the noncoexistence of anthocyanins and betalains were proposed, which are shown in Figure 7. First, in species where only betalains are present, the complete anthocyanin biosynthesis pathway might have been lost at some steps or was not yet developed (Figure 7A) [1,79]. As shown for the biosynthesis route of anthocyanins in Figure 5, there are many key important enzymes for the normal biosynthesis of anthocyanins, such as chalcone synthase and anthocyanin synthase. Similarly, the biosynthesis pathway of betalains might be blocked or incomplete in anthocyanin-producing plants (Figure 7B), although this has not been determined experimentally. Some studies indicate that MYB, as a cofactor of regular betalain biosynthesis in beet, is better than MYB in the anthocyanin pathway during evolution because MYB does not interact with bHLH or WD40 members of heterologous anthocyanin MBW complexes in beet [27], which implies a large evolutionary event allowing betalain to largely functionally replace anthocyanin [27]; that is, anthocyanin evolved into betalain. In addition, if the biosynthesis pathways for the two pigments in the same species were both intact, plants would have the ability to synthesize both pigments in theory. Perhaps because the biosynthesis of betalains consumes relatively more material and energy, the synthetic pathway always evolved toward the synthesis of fast, efficient and anthocyanins, which were consumed in low amounts, similar to a short circuit. However, this idea has not been proven. Regardless, all of these possibilities require further experimental verification.

## 6. Conclusions and Outlook

From the abovementioned studies provide a summary of the advances, developments and applications in the field of betalains. It is known that even though there have been many studies on betalains, their application in the stress response is not well understood. While there are quite a number of studies on the role of anthocyanins in plant response to stresses, through comparison of many researches, and we can think that betalain is more helpful for plants to adapt to adverse environment. As regards the process of evolution, betalain may be earlier than anthocyanin, further determination is yet to be explored.

Faced with human needs for resources such as pigments, energy and food, betalain biosynthesis-related genes may be a series of important, alternative and potential pigment genes, such as the genes related to DOD and cytochrome P450 series, with the help of genetic engineering technology [74], which are worthy of further investigation. Some studies have produced betalain through the callus that produces betalain, which opens a door for the rapid and large production of natural bioactive betalain and might be benefit for the food, pharmaceutical and cosmetic industries [15]. Some study can produce betalain in *Arabidopsis* transformed with the gene from betalain biosynthesis pathway, such as *BvMYB1* [27], even just given a single enzyme (DOD) and a enzyme’s substrate (L-DOPA) [82], which not only provide the possibility of enhancing the resistance of non-betalain-producing species, but also opens up new ways to produce betalain.

## Figures and Tables

**Figure 1 molecules-24-03078-f001:**
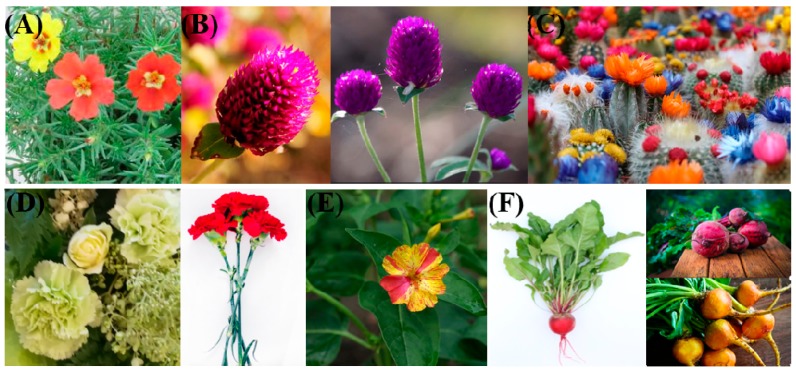
Several plants containing betalains. (**A**) *Portulaca oleracea* L. flowers (**B**) *Gomphrena globosa* L. flowers (**C**) *Echinopsis tubiflora* flowers (**D**) *Dianthus caryophyllus* L. flowers and stems (**E**) *Mirabilis jalapa* L. flowers (**F**) *Beta vulgaris* L. whole plants (some pictures from pexels.com).

**Figure 2 molecules-24-03078-f002:**
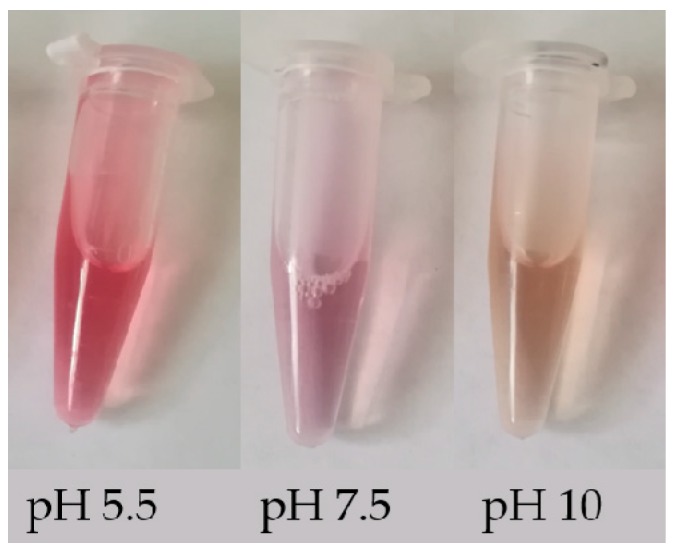
Color change of betalain in solutions with different pH value.

**Figure 3 molecules-24-03078-f003:**
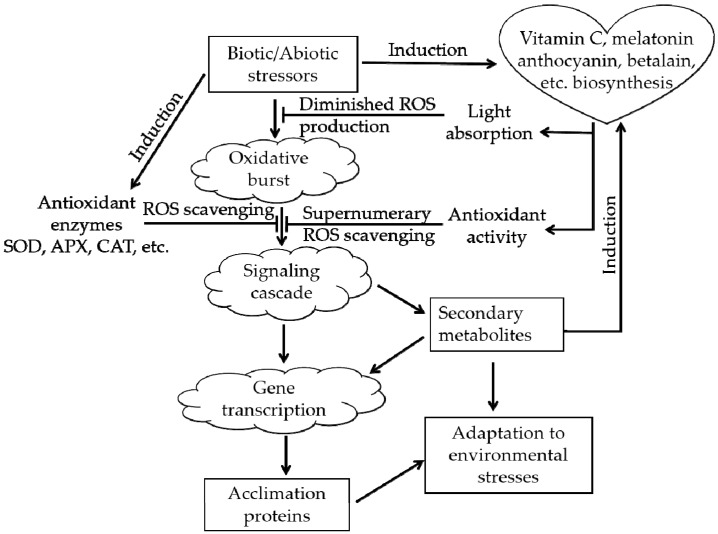
The roles of anthocyanins and betalains in the reactive oxygen species (ROS)-scavenging system (adapted from [31,32]). The arrowheads and T-bars indicate activation and suppression, respectively.

**Figure 4 molecules-24-03078-f004:**
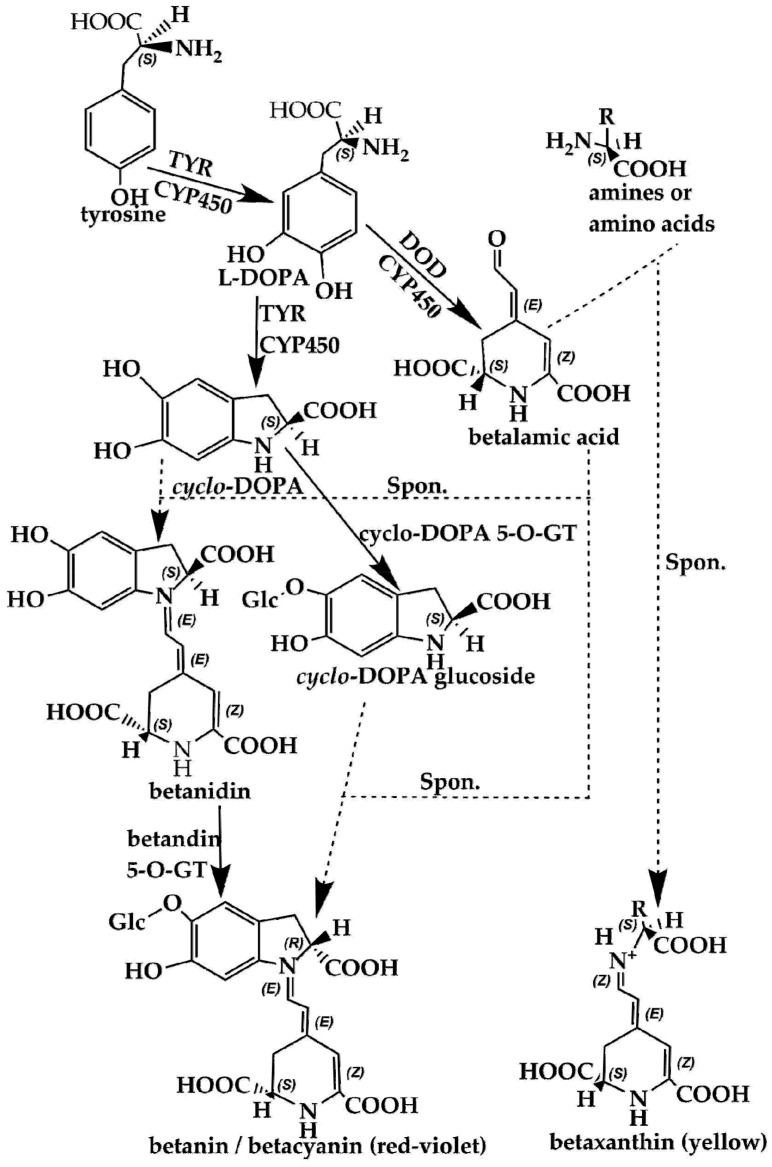
Biosynthesis pathways of betalains showing the main steps and intermediates. TYR—tyrosinase including tyrosinase hydroxylase and 5,6-dihydroxy-phenylalanine (DOPA)-oxygenase activity; DOD—DOPA 4,5-dioxygenase; spon.—spontaneous condensation reaction steps; GT—glycosyl transferase; DOPA—5,6-dihydroxy-phenylalanine; CYP450—cytochrome P450, e.g., CYP76AD1, CYP76AD6.

**Figure 5 molecules-24-03078-f005:**
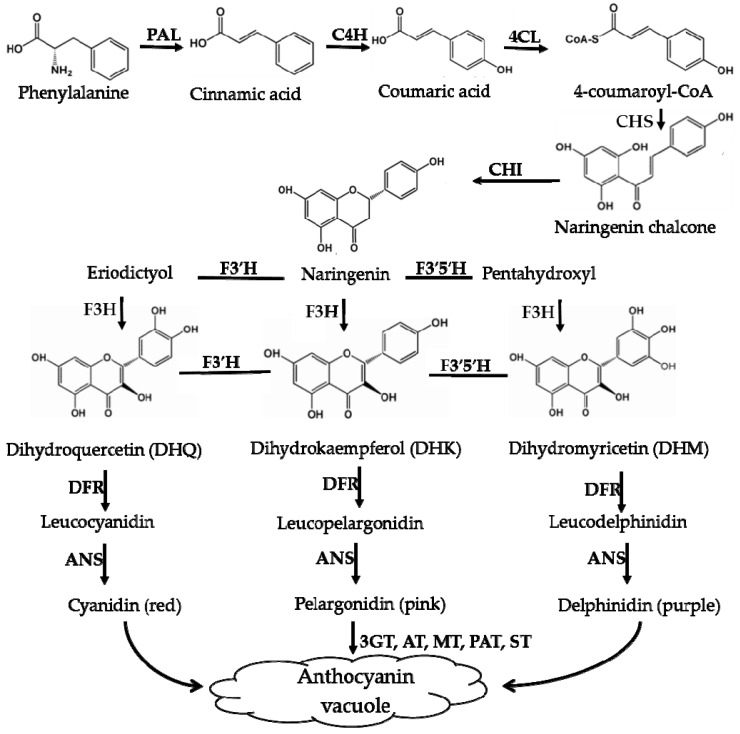
Biosynthesis pathways of anthocyanins. PAL—phenylala-nine ammonialyase; C4H—cinnamate-4-hydroxylase; 4CL—4-coumaroyl:CoA-ligase; CHS—chalcone synthase; CHI—chalcone isomerase; F3′H—flavonoid-3′ -hydroxylase; F3′5′H—flavonoid-3′,5′-hydroxylase; DFR—dihydroflavonol-4-reductase; ANS—anthocyanin synthase; F3H—flavonone-3-hydroxylase; F5H—ferulate 5-hydroxylase.

**Figure 6 molecules-24-03078-f006:**
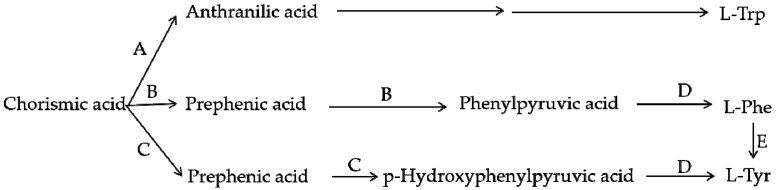
Branch pathway of aromatic amino acid biosynthesis. A—Anthranilate synthase; B—Chorismate mutase P-prephenate dehydrase; C—Chorismate mutase T-prephenate dehydrogenase; D—transaminase; E—phenylalanine hydroxylase.

**Figure 7 molecules-24-03078-f007:**
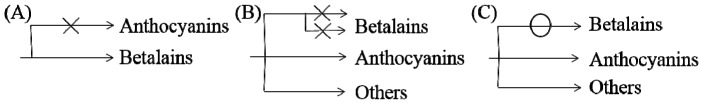
Possible mechanisms of noncoexistence of anthocyanins and betalains. (**A**) only betalains are present and anthocyanin biosynthesis pathway might be incomplete in betalain-producing plants (**B**) biosynthesis pathway of betalains may be blocked or incomplete in anthocyanin-producing plants (**C**) biosynthesis pathway of betalains may be may be short-circuited in anthocyanin-producing plants.

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
