# Peer review of "Research Progress of Betalain in Response to Adverse Stresses and Evolutionary Relationship Compared with Anthocyanin"

_molecules, 2019, doi:10.3390/molecules24173078_

Round 1

Reviewer 1 Report

The objetive of this review was summarize the research progress on betalain responses to  adverse stresses comparing with anthocyanin.

The topic is relevant to the scope of Molecules. However, in my opinion it requires a major revision before be suitable for publication, addressing the following aspects:

In the introduction, I suggest the authors to include more information’s about the previous works performed about:

-Biological activies of plant pigments betalains.

-I suggest including ranges of concentrations of both pigments, betalains and anthocyanins, in plants under different adverse conditions (metal toxicity, , drought, salinity, . The review lacks data.

Line 69-71. This paragraph must be rewritten, is not clear.

The manuscript do not objectively reflect the most significant findings in the Conclusion section. Also, the authors  should explain better the challenges and opportunities about the future research in these pigments.

Reviewer 2 Report

The manuscript “Research Progress on Betalain Responses to Adverse Stresses Comparing with Anthocyanin” is a quite comprehensive review about the roles of betalains and anthocyanins in plants (and humans), their metabolic biosynthetic pathways, and evolutionary relationships. The topic is of course interesting, but I think the manuscript needs a better organization in its structure. Some parts are indeed redundant and others should be improved. Here below my comments are reported:

1)   Title: the title should be changed because not exhaustive of all the parts composing the review.

2)   Page 2, Identification and detection of betalains: in this part, the identification of betalains performed with more advanced techniques is not mentioned (e.g. HPLC, HPLC-MS), but must be included.

3)   Line 84: Please, re-write this sentence because as it is, is not clear.

4)   Page 4, Comprehensive effects of various stresses on plants in the environment: This part is the main concerning, because there is a wide description of ROS and antioxidant systems, but only one sentence at the end focused on betalains. In my opinion, this part should be avoided or incorporated in paragraph 3 ( 3.1). The font is sometimes different

5)   Line 109: this sentence is the same at line 81.

Round 2

Reviewer 1 Report

Accept in present form

This manuscript is a resubmission of an earlier submission. The following is a list of the peer review reports and author responses from that submission.

Round 1

Reviewer 1 Report

The article has slightly improvement compared to the previous version, but it is still extremely hard to follow both in language and the logic that organize the entire review.

Take the abstract for an example, 

line11,  the word choice "thoroughly", it is "thoroughly" study, why are still still so many understanding questions? 

lline14, adversity-> adversarial

ikewise, line 19 "complete", what does "the physiological evolution of betalines is almost complete" mean? The evolutionary relationship between betalains and anthocyanins is a complicated one in terms the evolutionary origin of each pathways. But putting the physiological evolution of betalines and evolutionary origin of each pathways in a single sentence make it impossible to understand what the authors are really trying to tell.

I would suggest the authors to find a native english speaker to thoroughly correct the entire manuscript. Sorry that I cannot provide a more positive comments on this review.

Reviewer 2 Report

The objetive of this review was summarize the research progress on betalain responses to  adverse stresses comparing with anthocyanin.

The manuscript is sound and has merits for publication in Molecules. However, in my opinion it requires a major revision before be suitable for publication, addressing the following aspects:

In the introduction, I suggest the authors to include more information’s about the previous works performed about:

- the potencial use of sources of betalain (funcional foods, …)

-Biological activies of plant pigments betalains

-I suggest including ranges of concentrations of both, betalains and anthocyanins, in plants under different adverse conditions (metal toxicity, , drought, salinity, . Maybe a table?

The manuscript do not objectively reflect the most significant findings in the Conclusion section. Also, the authors  should explain better the challenges and opportunities about the future research in these pigments.